# The numerical simulation of droplet impact on surfaces is conducted using the SPH-DEM method

**Shilong Bu**[1]*, **Daming Li**[2], **Hu Tao**[1], **Wenjie Hou**[1]

**1** School of Civil Engineering, Lanzhou Jiaotong University, Lanzhou, China, **2** State Key Laboratory of Hydraulic Engineering Simulation and Safety, Tianjin University, Tianjin, China

* busl@lzjtu.edu.cn

## Abstract

The process of liquid droplet impinging upon the surface of particles entails complex dynamics and significant deformation. In this study, the smoothed particle hydrodynamics (SPH) method coupled with the discrete element method (DEM) is employed to investigate the motion process of liquid droplet impacting the particle surface. A surface tension model is introduced into the SPH motion equation to calculate the motion of the liquid droplet. In the SPH-DEM coupling module, the viscous force and capillary force between the liquid droplet and the particles are taken into account. The surface tension model is verified through two cases: the free deformation process of a stationary square liquid droplet and the impact of a liquid droplet on a hydrophobic wall. The accuracy of the DEM model is validated through experimental verification of dry particle collapse. And the experimental results validate the accuracy of the SPH-DEM model in simulating the liquid droplet impact on the particle surface. The simulation results are in good agreement with the experimental ones. Utilizing the SPH-DEM model, the influences of the droplet impact velocity and the particle diameter on the rebound phenomenon after the water droplet impacts the particle wall of the powder bed are respectively investigated. The results indicate that the higher the droplet impact velocity and the smaller the diameter of the powder bed particles, the faster the rebound rate of the droplet after impacting the powder bed layer.

## 1. Introduction

The collision between droplets and particles is a common occurrence in nature, daily life, as well as industrial and agricultural production. For instance, raindrop impact on sand [1], high-speed water flow for dust pollution cleaning in industrial processes [2], binder impact on particle surfaces during wet granulation [3], and the formation process of composite materials [4]. The interaction between droplets and particles upon impact is a complex coupled process involving gas, liquid, and solid multiphase

**Data availability statement:** We have uploaded the data into an online database, which can be found at the link below. https://doi.org/10.6084/m9.figshare.28659227

**Funding:** The work was supported by the National Natural Science Foundation of China (No. 52360031), Gansu Province higher education industry support plan project (No. 2022CYZC-32), and The Young Scholars Science Foundation of Lanzhou Jiaotong University (No. 1200061253). The funders had no role in study design, data collection and analysis, decision to publish, or preparation of the manuscript.

conditions. During this process, the behavior of droplets is influenced by various factors such as impact force and surface tension. Researchers are particularly interested in understanding the movement of particles after being impacted by droplets, deformation patterns, and the interaction between moving droplets and particles [5].

The characteristics of droplets (nature, size, initial velocity) as well as particle properties within the bed significantly influence agglomeration resulting from droplet-particle impacts [5–8]. Although experimental methods have been extensively employed to study the phenomenon of droplet-particle collisions [9–11], limitations regarding time scale and space scale make it challenging to obtain detailed information about internal velocity fields, force fields, or multiphase media interactions. Consequently, many scholars have turned their attention towards numerical simulations which hold great potential for providing more comprehensive insights.

A droplet impacting the particle surface is a complex coupled action process under multiphase conditions involving gas, liquid, and solid. To gain a deeper understanding of this process, computer simulation can be an effective tool as it has the potential to provide detailed information that is challenging to obtain experimentally [12]. Most macroscopic models [13] primarily focus on obtaining the flow field and flow characteristics. In mesoscale simulation [12], the average volume equation is typically solved using a mesoscale grid model to acquire a representative flow field, where the grid cell size exceeds that of the particle size. Microscale models [14] are capable of capturing fluid effects and individual particles' behavior. During phase interface determination, the accuracy of volume-of-fluid (VOF) method simulations depends on the scale of the model grid [15]. The mesh method simulates collision processes between different droplets, allowing for reproduction of droplet merging and separation; however, it cannot replicate droplet bouncing due to considering two droplets' surfaces as merged when they enter the same grid in phase interface treatment methods [16]. This necessitates developing new boundary conditions to address this limitation [17]. The Level Set method describes phase interfaces through zero contours of high-dimensional functions without requiring explicit tracking, making it easily extendable to any dimensional space and efficiently handling complex material interfaces and their changing topological structures [18].

The advantage of these methods lies in the direct calculation of surface tension from the tracked interface shape. It is important to note that these models require intensive mesh recalculation in order for the mesh to conform accurately to the interface shape, resulting in high computational cost and decreased accuracy. The development of meshless methods provides a solution to this problem. The Smoothed Particle Hydrodynamics (SPH) method was initially proposed by Lucy, Gingold, Monaghan, and others in 1977 as a meshless approach with distinct advantages in handling large deformation problems involving free surfaces [19,20]. Due to oscillations before collision with solid surfaces, droplets typically exhibit irregular shapes. The SPH method has extensively investigated droplet vibration and deformation under various external forces and surface tensions [21–24]. In recent years, droplet impact simulation methods have achieved remarkable progress in multi-physics field coupling and complex interface handling. For instance, He et al. [25] revealed the aggregation effect of liquid metal droplets on elastic substrates through SPH method, providing novel insights into

energy dissipation mechanisms at soft material interfaces; Dong et al. [26] developed a dual-core SPH model that effectively suppresses tensile instability, laying methodological foundations for contact time regulation of macrostructurally structured surfaces; Meanwhile, Huo et al. [27] comparatively demonstrated the robustness of Lagrangian framework-based fluid-solid coupling simulations by contrasting FEM-SPH and SPH-SPH approaches in elastic beam vibration analysis.

While extensive experimental investigations have demonstrated droplet retraction upon contact with particulate surfaces, the factors governing the rate of this phenomenon remain insufficiently investigated. Therefore, this study proposes the SPH-DEM coupled model to investigate droplet retraction, which considers both the viscous force between droplets and particles and the capillary force caused by voids. The coupling model code was implemented in Fortran 95 language using Visual Studio 2010 and Inter Visual Fortran 2010 as the compilation environment. The model adopts a meshless method for interface tracking. The validity of the proposed coupling model in this study is verified through relevant research literature. Based on this, the effects of droplet impact velocity and particle diameters on droplet retraction rate were investigated using the model.

## 2. Methodology

### 2.1 SPH method

The SPH method is a meshless method for solving the governing equations based on Lagrangian particles. The basic equation of SPH is:

$$A(\mathbf{r}_i) = \sum_j A(\mathbf{r}_j) W(\mathbf{r}_i - \mathbf{r}_j, h) \frac{m_j}{\rho_j}$$

(1)

where $A$ can be any function of the fluid field, $j$ is the particle in the neighborhood of particle $i$; $\mathbf{r}_i$ and $\mathbf{r}_j$ are the positions of particle $i$ and $j$, respectively; $W(\mathbf{r}_i-\mathbf{r}_j,h)$ is called the kernel function or smoothing function; $m$ and $\rho$ are the mass and density, respectively. The size of the neighborhood depends on the smoothing length h of the kernel function $W$. In the present study, the cubic spline kernel function is applied:

$$W(r, h) = \alpha_d \begin{cases} (2-q)^3 - 4(1-q)^3, & 0 \le q < 1 \\ (2-q)^3, & 1 \le q < 2 \\ 0, & q \ge 2 \end{cases}$$

(2)

where $q = r/h$; $r$ is the distance between particles $i$ and $j$:$h$ is the smoothing length:$\alpha_d$ is the regularization coefficient $\alpha_d = 5/14\pi h^2$ in two-dimensional.

The governing equations of fluid motion are as follows,

$$\frac{d\rho}{dt} = -\rho \nabla \cdot \mathbf{v}$$

(3)

$$\frac{d\mathbf{v}}{dt} = \mathbf{g} - \frac{1}{\rho}\nabla p + \frac{\mu}{\rho}\nabla^2 \mathbf{v}$$

(4)

where $\mathbf{v}$ is the fluid velocity, $p$ is the fluid pressure, $\mu$ is the dynamic viscosity of the fluid, and $\mathbf{g}$ denotes the body force acting on the fluid, such as gravitational force. With the SPH kernel summations, the governing equations can be written in the following form,

$$\frac{d\rho_i}{dt} = \sum_{j=1}^{N} m_j \mathbf{v}_{ij} \cdot \nabla_i W_{ij}$$

(5)

$$\frac{d\mathbf{v}_i}{dt} = \mathbf{g}_i - \sum_j m_j \left( \frac{P_i}{\rho_i^2} + \frac{P_j}{\rho_j^2} \right) \nabla_i W_{ij} + \sum_j \frac{2m_j \mu \mathbf{r}_{ij} \cdot \nabla_i W_{ij}}{\rho_i \rho_j \left( r_{ij}^2 + \eta \right)} \mathbf{v}_{ij} \tag{6}$$

where $\mathbf{v}_{ij} = \mathbf{v}_i - \mathbf{v}_j$, $\mathbf{r}_{ij} = \mathbf{r}_i - \mathbf{r}_j$, $r_{ij} = |\mathbf{r}_{ij}|$. $h$ is the smoothing length. The third term on the right side of the Eq. (6) is the viscous term. The term $\eta = 0.01h^2$ was added to prevent the singularity of the viscous term when two particles approach each other infinitely.

The artificial viscosity demonstrates notable advantages in complex free-surface scenarios such as wave breaking and splash [28–30]. In this study, however, the physical viscous scheme was adopted to prioritize numerical stability and direct alignment with the Navier-Stokes equations.

In the context of mesoscopic and microscopic fluid dynamics, surface tension plays a crucial role that cannot be disregarded. Commonly employed models for surface tension include those based on continuum surface force (CSF) and the inter-particle interaction force (IIF) model proposed by Tartakovsky et al. [31]. In this study, we incorporate the Van der Waals stress term into the equation governing fluid motion to simulate surface tension. This approach avoids direct application of forces between particles and maintains higher consistency with the equation governing fluid motion. After considering the influence of surface tension, the motion equation is modified as follows:

$$\frac{d\mathbf{v}_i}{dt} = \mathbf{g}_i - \sum_{j=1}^{n} m_j \left( \frac{P_i}{\rho_i^2} + \frac{P_j}{\rho_j^2} \right) \nabla_i W_{ij} + \sum_j \frac{2m_j \mu \mathbf{r}_{ij} \cdot \nabla_i W_{ij}}{\rho_i \rho_j \left( r_{ij}^2 + \eta \right)} \mathbf{v}_{ij} - \sum_{j=1}^{N} m_j \left( \frac{\tilde{p}_i}{\tilde{\rho}_i^2} + \frac{\tilde{p}_j}{\tilde{\rho}_j^2} \right) \nabla_i \tilde{W}_{ij} \tag{7}$$

where $\tilde{p}$ represents the van der Waals stress; $\tilde{\rho}$ denotes the van der Waals density.

$$\tilde{p} = -k\tilde{\rho}^2 \tag{8}$$

$k$ represents the van der Waals stress.

In this study, the Van der Waals form of the equation of state is used in the study of surface tension [32]:

$$P = \frac{\rho \bar{k} T}{1 - \rho \bar{b}} - \bar{a} \rho^2 \tag{9}$$

where $\bar{k} = k_b/m$ ($k_b$ is the Boltzmann constant), $\bar{a} = a/m^2$, and $\bar{b} = b/m$. Here, $a$ and $b$ are the Van der Waals constants and $m$ is the mass of the particles. $m_i = \rho_i V_i$, $\rho_i$ represents the density of the particle, and $V_i$ represents the volume of the particle.

In order to avoid the non-physical oscillation caused by the lack of particles near the wall boundary of the support domain, the pressure correction method is adopted, and the pressure field is updated every 30 timesteps [33].

## 2.2 DEM method

The Discrete Element Method (DEM) was first proposed by Peter Cundall [34] in 1971 as a numerical method for simulating the movement of solid particles in discontinuous media. DEM adopts Newton's second law of motion to update each particle's velocity and position, and force-displacement law to calculate each particle's contact force exerted by surrounding particles or boundaries. The equation of motion of particle $\alpha$:

$$m_\alpha \frac{d\mathbf{u}_\alpha}{dt} = \mathbf{F}_\alpha \tag{10}$$

$$I_\alpha \frac{d\omega_\alpha}{dt} = \mathbf{M}_\alpha \tag{11}$$

where $m_a$, $\mathbf{u}_a$ and $\mathbf{F}_a$ are the solid particle's mass, translational velocity, and the sum of particle–particle interaction forces, respectively; $\omega_\alpha$ is the solid particle's rotational velocity and $\mathbf{M}_a$ is the resultant moment, respectively; is the moment of inertia.

According to the force synthesis principle, the final resultant force and moment received by particle $i$ are:

$$\mathbf{F}_\alpha = \sum_{\beta=1}^{k} (\mathbf{fn}_{\alpha\beta} + \mathbf{ft}_{\alpha\beta}) \tag{12}$$

$$\mathbf{M}_\alpha = \sum_{\beta=1}^{k} (\mathbf{M}^t_{\alpha\beta} + \mathbf{M}^r_{\alpha\beta}) \tag{13}$$

where $k$ is the number of particles $\beta$ in contact with particle $a$ in the computational domain; $\mathbf{fn}_{\alpha\beta}$ is the normal contact force between particles $\alpha$ and $\beta$; $\mathbf{ft}_{a\beta}$ is the tangential contact force between particles $\alpha$ and $\beta$; $\mathbf{M}^t_{\alpha\beta}$ is the tangential torque on particle $a$ due to contact with particle $\beta$; $\mathbf{M}^r_{\alpha\beta}$ is the rolling resistance torque on particle $a$ due to contact with particle $\beta$.

The damping effect between particles reflects the dissipation of energy during the collision process. Considering the damping effect, the force on the particles can be expressed as:

$$\begin{cases} \mathbf{fn} = -(k_n\lambda + c_n\mathbf{v}_{ij} \cdot \mathbf{n_n})\,\mathbf{n_n} \\ \mathbf{ft} = -(k_t\delta + c_t\mathbf{v}_{ij} \cdot \mathbf{n_t})\,\mathbf{n_t} \end{cases} \tag{14}$$

where $k_n$ is the normal spring stiffness of linear spring; $k_t$ is the tangential spring stiffness of linear spring; $\lambda$ and $\delta$ are the normal and tangential overlap, respectively; $\mathbf{v}_{a\beta}$ is the relative velocity between particles $\alpha$ and $\beta$ at contact point; $\mathbf{n_n}$ and $\mathbf{n_t}$ ate the normal and tangential unit vector pointing from particle $\alpha$ to particle $\beta$, respectively.

Based on the Hertz-Mindlin contact theory [35], the expression of $c_n$ is:

$$c_n = -2\xi_n\sqrt{k_n m_{\mathrm{eff}}} \tag{15}$$

where $m_{\mathrm{eff}} = \frac{m_i m_j}{m_i + m_j}$ represents the equivalent mass. $\xi_n$ is the normal damping ratio, calculated through the recovery coefficient e:

$$\xi_n = \frac{-\ln e}{\sqrt{\pi^2 + (\ln e)^2}} \tag{16}$$

The tangential damping coefficient $c_t$ is generally proportional to the normal damping.

$$c_t = \eta \cdot c_n \tag{17}$$

The scaling factor $\eta \in [0.5 \sim 1.0]$ [36].

According to the Hertz–Mindlin theory [35], the expression of the normal spring stiffness coefficient $k_n$ is:

$$k_n = \frac{4}{3}\left(\frac{1-\nu_i^2}{E_i} + \frac{1-\nu_j^2}{E_j}\right)^{-1}\left(\frac{r_i + r_j}{r_i r_j}\right)^{-1/2} \tag{18}$$

where $E$, $\nu$, $r$ are the Young's modulus, Poisson's ratio and radius of the particle, respectively.

The expression of the tangential spring stiffness coefficient $k_t$ is:

$$k_t = 8\lambda^{1/2}\left(\frac{1-\nu_i^2}{G_i} + \frac{1-\nu_j^2}{G_j}\right)^{-1}\left(\frac{r_i+r_j}{r_ir_j}\right)^{-1/2}$$

(19)

where $G$ is the shear modulus of the particle.

If the two particles are not in contact, then both normal and shear contact forces are set to zero according to the criterion of no normal strength in tension. Otherwise, calculating the maximum allowable shear contact force $ft_{max}$ according to Coulomb friction law:

$$ft_{max} = \mu fn$$

(20)

where $\mu$ is the smaller friction coefficient of the two contact particles.

If $|ft| > ft_{max}$, that is, when the magnitude of the tangential contact force exceeds the maximum allowable tangential contact force, the tangential contact force is reduced to the limit value while retaining its sign:

$$ft = ft\frac{\mu fn}{|ft|}$$

(21)

## 2.3 SPH-DEM coupling module

The interaction between particles in the SPH method is typically achieved by interpolating the physical eigenvalues using a kernel function. In contrast, the DEM method relies on direct contact deformation processes between particles combined with Newton's laws of motion to facilitate particle interactions.

On the mesoscopic scale, there exist distinct gaps between solid particles. Due to capillary action, water infiltrates these boundaries and is drawn into the interstices of the particles, resulting in an indirect adsorption force $f^m$ between the solid particles and water. By calculating the repulsive force, it becomes possible to interpolate the pressure on solid particles and subsequently substitute it into the fluid motion equation for solving forces and accelerations. In practical applications, particle profiles are intricate, particle motions are uncertain, and particle materials are non-uniform. In this study, we simplify the adsorption force caused by capillary action as a direct force acting between fluid particles and solid particles. Assuming that voids within a small range near solid particles are interconnected, these voids can be treated as a whole during calculations. Similar to fluid particles in SPH (Smoothed Particle Hydrodynamics), support domains can be assigned to solid particles in DEM (Discrete Element Method), enabling determination of saturation $S_a$ [37] by interpolating both the number of solid particles and fluid particles within each particle's support domain. The adsorption force induced by capillary action is influenced by both wetting coefficient $\varsigma$ [38]and pore pressure $P$ in proximity to each particle.

$$\mathbf{f}_i^m = -\varsigma_i\sum_{\beta=1}^{NL} m_i m\beta\frac{p_\beta}{\rho_i\rho_\beta}\nabla_\beta W$$

(22)

where $NL$ represents the number of fluid particles in the support domain of solid particle $\beta$; $\varsigma$ is the wetting coefficient of the particles, which reflects the infiltration ability of the fluid to the granular particles. Here, $\varsigma$ is related to the particle material property coefficient

$\varsigma'$ and the degree of saturation $Sa$ in the vicinity of the particle.

$$\varsigma = \left( \frac{1}{Sa} - 0.5 \right) \varsigma' \tag{23}$$

$$Sa = \sum_{\beta=1}^{NL} \frac{m_\beta}{\rho_\beta} W_{i\beta} \tag{24}$$

To sum up, the force $\mathbf{f}_i^c$ of the fluid particle $i$ on the surrounding solid particle $\beta$ in the analytical coupling model should include three parts, viscous shear force $\mathbf{f}^{ten}$, adsorption force $\mathbf{f}^m$, and repulsion force $\mathbf{f}^{rep}$:

$$\mathbf{f}_i^c = \sum_{\beta=1}^{NS} \left( \mathbf{f}_\beta^{rep} - \varsigma_\beta m_i m_\beta \frac{p_i}{\rho_i \rho_\beta} \nabla_i W_{i\beta} + m_i m_\beta \left( \frac{4 \upsilon \mathbf{x}_{i\beta} \cdot \nabla_i W_{i\beta}}{(\rho_i + \rho_\beta) r_{i\beta}^2} \right) \mathbf{v}_{i\beta} \right) \tag{25}$$

where
$\upsilon$ is the kinematic viscosity coefficient of the fluid.

Similarly, the force $j$ is subjected to the surrounding solid particle $i$ can be expressed as:

$$\mathbf{f}_\beta^c = \sum_{i=1}^{NL} \left( \mathbf{f}_i^{rep} + \varsigma_i m_i m_\beta \frac{p_i}{\rho_i \rho_\beta} \nabla_i W_{i\beta} + m_i m_\beta \left( \frac{4 \upsilon \mathbf{x}_{i\beta} \cdot \nabla_i W_{i\beta}}{(\rho_i + \rho_\beta) r_{i\beta}^2} \right) \mathbf{v}_{\beta i} \right) \tag{26}$$

The coupled motion equations can be obtained by bringing the interaction force between solid particles and fluid particles into the respective motion equations of the SPH and DEM models.

## 2.4 Time Step

The selection of the time step significantly impacts both the stability and efficiency of the model. The critical time step in the DEM model should be associated with the propagation speed of Rayleigh Waves along the surface of solid particles [39]. It is assumed that when two circular particles with a radius R collide, only a Rayleigh Wave propagates between them without affecting other particles (this assumption is similar to that in the DEM soft sphere model). Furthermore, it is assumed that particle interactions are limited to those occurring between contacting particles. Therefore, for this model, the critical time step should be less than or equal to the time required for Rayleigh waves to propagate along a hemisphere [40],

$$\Delta t_{crit} = \frac{\pi R_p}{0.1631\nu + 0.877} \sqrt{\frac{\rho}{G}} \tag{27}$$

where
$\triangle t_{crit}$ is the critical time step; $R_p$ is radius;
$\nu$ is Poisson's ratio;
$G$ is the shear modulus of the solid particulate material.

The SPH method uses an explicit calculation mode and estimates the time step with the CFL condition. According to the Courant-Friedrichs-Lewy (CFL)condition, the maximum value of the numerical transfer velocity should be larger than the maximum value of the material transfer, so the size of the critical time step in the SPH model should be proportional to the distance between particles. Morris [41] proposed the expression of critical time steps considering viscous diffusion.

 

$$t_{\text{crit}} \leq \lambda \min \sqrt{\frac{h_i}{|f_i|}} \qquad (28)$$

$$t_{\text{crit}} \leq 0.125 \frac{h_{\min}^2}{\upsilon} \qquad (29)$$

Where $\lambda$ is the safety factor, $\lambda = 0.25$ [7]; $f$ is the acceleration of the particle; $\upsilon$ is the kinematic viscosity coefficient of the fluid.

According to the critical time steps $t_{\text{crit}}^L$ and $t_{\text{crit}}^S$ of the SPH and DEM models, the relationship between the two-time steps can be obtained:

$$rt = \text{int}\left(\frac{\max\left(t_{\text{crit}}^L\right)}{\min\left(t_{\text{crit}}^S\right)} + 1\right) \qquad (30)$$

where $rt$ is the time step control parameter of the coupled model.

## 3 The results of numerical simulation

### 3.1 The test of square drop oscillation

The present example simulates the free deformation process of a stationary square droplet under vacuum conditions, serving as validation for the surface tension model. In this simulation, 2500 droplets with an initial particle spacing of $\Delta x = 0.04$ mm are placed in a vacuum environment, and their size is set to be 2 mm × 2 mm. These droplets are allowed to deform freely under the influence of surface tension until they form circular shapes. The surface tension value used in this simulation is 0.02361 N/m. A time step of $dt = 1.0 \times 10^{-4}$ ms is employed, along with a dynamic viscosity coefficient $\eta = 1.0 \times 10^{-3}$ Pa·s and an initial density of fluid particles $\rho = 1000$ kg/m³. Additionally, the smooth length $h$ is defined as 1.3 times $\Delta x$ ($h = 1.3\Delta x$). Fig 1 illustrates the different methods used to capture the shape evolution of these droplets at various time points [42,43]. It can be observed that all methods yield circular shapes for the final droplet configuration; however, this study proposed method achieves greater symmetry compared to others, which aligns more closely with actual physical phenomena occurring in gravity-free environments.

### 3.2 Droplet impact on the hydrophobic wall

In this section, a numerical simulation is conducted to investigate the motion of a droplet impacting a superhydrophobic surface at a velocity of $v = 0.54$ m/s, taking into account the combined effects of gravity, viscosity, and surface tension. The simulation results are then compared with experimental data from Li et al. [44] for validation purposes. For this specific case study, the droplet has a radius of $R = 1.088$ mm, and the calculation utilizes a time step of $dt = 1.0 \times 10^{-4}$ ms. The dynamic viscosity coefficient is $\eta = 1.0 \times 10^{-3}$ Pa·s, while the initial density of fluid particles is $\rho = 1.0 \times 10^3$ kg/m³. Additionally, the acceleration due to gravity is $g = 9.8$ m/s² and the van der Waals attractive coefficient is $k = 9.66 \times 10^{-3}$. The simulated results are presented in Fig 2a for comparison with an experiment conducted by Li where they observed an impact between a droplet traveling at $v = 0.54$ m/s and a solid superhydrophobic material surface as shown in Fig 2b.

The droplet descends due to inertia and upon reaching the wall, it spreads along its surface as depicted in Fig 2 from 0–1.8 ms. A portion of the droplet's kinetic energy is converted into potential energy, while the remaining dissipates due to viscosity. As the droplet's surface area increases, the resistance of surface tension to spreading becomes more evident until the droplet's kinetic energy reaches zero. At this point, the contact area between the droplet and solid wall reaches its maximum; owing to surface tension's "pulling back" effect, as shown in Fig 2 at 4.4 ms, the edge thickness exceeds that of the center.

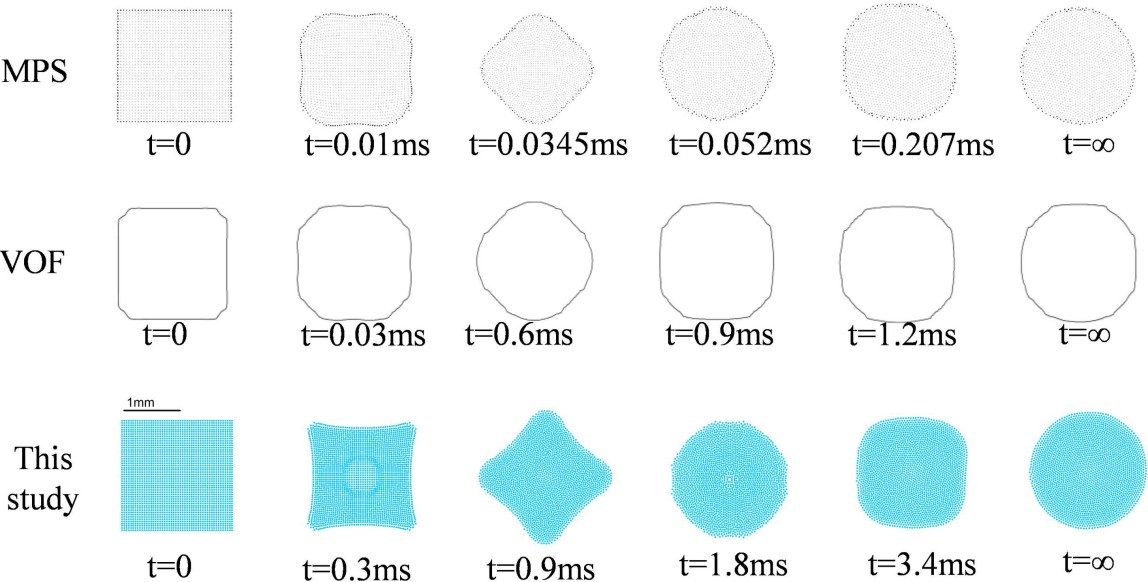

**Fig 1. Simulation of the deformation process of a square droplet under the action of surface tension.**

Governed by dominant surface tension, the droplet's surface area continues to decrease until its contact with solid wall diminishes completely resulting in rebound and separation from it as depicted in Fig 2 from 9.4–14.8ms. Overall, simulation results in this section are consistent with Li's experimental findings; comparison figures indicate that apart from discrepancies during rebounding stage, simulation outcomes during impact and retraction stages closely align with experimental results.

The Fig 3 depicts the temporal evolution of the wall-wetting diameter (*wd*) in both experimental data and simulation results. According to the simulation data, the time interval from 0 to 3.3ms corresponds to the droplet's spreading process on the solid wall surface. During this phase, the droplet demonstrates a higher spreading velocity compared to that observed in experimental data, and its maximum wetting diameter (*wd*) also exceeds the experimental value. The subsequent period from 3.3ms to 3.8ms represents the retraction process of the droplet, during which its retraction speed surpasses that of the experiment, with a greater disparity between them than observed during spreading. From a force analysis perspective, it is suggested that an excessively high movement speed of the droplet may be attributed to insufficient resistance against motion.

### 3.3 Validation of the DEM Model

In order to validate the accuracy of the DEM model established in this paper, a simple sand column collapse experiment was conducted and compared with the simulation results of the DEM model. The cross-section of the two-dimensional sand column model is rectangular, with a length (*L*) of 12 cm and a height (*H*) of 6 cm. As shown in Fig 4(a), it depicts the initial distribution of particles in the DEM model. The experimental fine sand used had a diameter ranging from 0.1mm to 0.5mm and a density of 2650 kg/m³. In the DEM model, particle diameter was set at 0.4mm, normal stiffness at $3.0 \times 10^7$N/m, tangential stiffness at $2.7 \times 10^7$N/m, Poisson's ratio at 0.3, damping coefficient at 0.04, recovery coefficient at 0.8, sliding friction coefficient at 0.4, and time step taken as $1.0 \times 10^{-5}$s. Fig 4(b) shows particle distribution after removing the retaining plate when the sand column collapsed to its final stable state; while Fig 4(c) presents an image capturing the stable state after collapse.

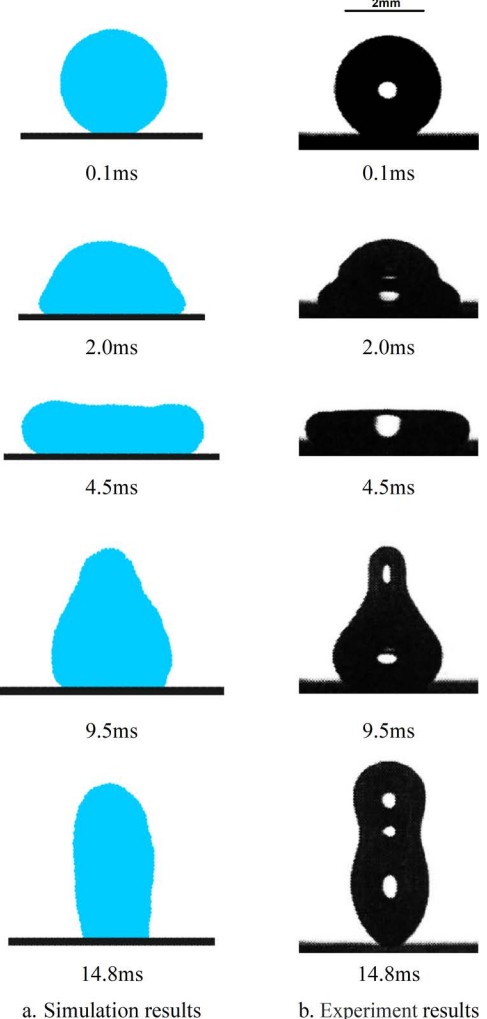

**Fig 2. Illustrates the dynamic process of a droplet impacting a hydrophobic wall surface at a velocity of v = 0.54m/s, presenting a comparison between simulation results and experimental data.**

Figs 5 and 6 illustrate the instantaneous distribution of particles and their velocities at time intervals $t=0.05s$, 0.3s, 0.5s, and 1.0s. Due to experimental constraints and measurement limitations, it was not feasible to accurately capture the instantaneous collapse of the particle column during the experiment; only the stabilized profile contour was obtained, precluding a direct comparison with simulated instantaneous conditions. Immediately after removing the baffle (as depicted in Fig 5), under gravity's influence, the sand column initially collapsed at its toe region forming a triangular sliding area at $t=0.05s$ while other particles remained relatively stationary. By $t=0.3s$, some particles had moved near the wall surface creating a sliding slope with a significant incline angle. At approximately $t=1.0s$, particle movement ceased resulting in a profile similar to that shown in Fig 4(c).

Fig 7 compared profiles formed by dry particle column collapse against those obtained from experiments. The line demarcating sliding particles from non-sliding ones is termed as relative rest line; the angle between this line and horizontal plane is referred to as relative rest angle. Simulated relative rest angle is about 60° while experimental value stands around 62°with less than5%relative error. Maximum runout distance from simulation measures 19.01 cm whereas experimental observation records 21.8 cm, resulting in an approximate 7.73% relative error.

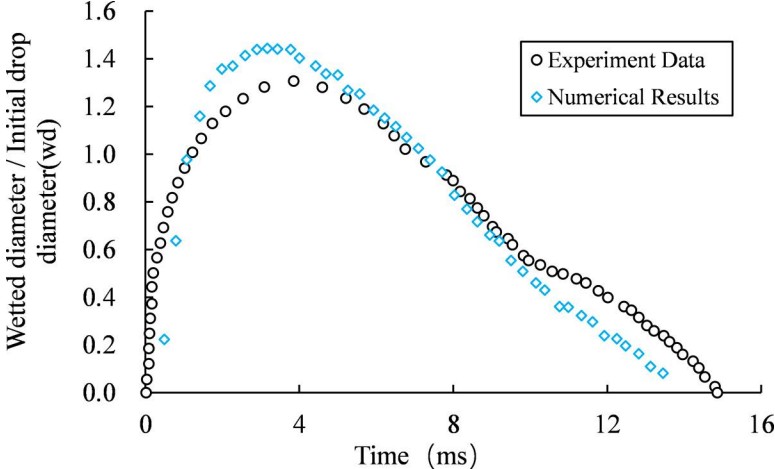

**Fig 3. The temporal evolution of wall wettability diameter, comparing simulation results with experimental data.**

### 3.4 Simulation of droplet impact on particle surface

The SPH-DEM model in this study is validated by the experimental observation of droplet impacting particle surface conducted by Marston et al. [45]. In Marston's experiment, water droplets with a diameter of $D_0 = 2.1$mm were used to impact particles with an average diameter of $d = 31$μm. The model parameter setting information can be found in Table 1.

The experimental snapshots of the droplet impacting the particle surface are compared with the simulated results obtained from the SPH-DEM model. In Fig 8, the gray background image represents high-speed camera recordings capturing the droplet movement during testing, while the colored image depicts the simulation results of the model. In this representation, blue corresponds to the droplet and red corresponds to the particles.

The droplet falls freely under the influence of gravity, and the initial moment of contact between the droplet and the particle surface is defined as $t = 0$ms. At $t = 1.5$ms, when the initial velocity of the droplet is 0.15m/s, it impacts the particle surface and spreads along both ends due to boundary resistance. The period from $t = 2$ms to 4.5ms represents the spreading process of droplets on the particle surface, where inertia force causes gradual spread of liquid in the middle part around. From $t = 4.5$ms to 6.5ms, retraction occurs due to surface tension acting on the droplet while certain particles are adsorbed on its contact surface through capillary action during spreading process. When the initial velocity of the droplet is 2.2 m/s, experimental and simulation results show a gradual decrease in thickness at its middle part with an increase at its edge due to pullback effect caused by surface tension on spreading fluid [46]. Fig 9 illustrates a plot depicting relationship between spread diameter of droplets on particle surface and impact time for two different velocities ($v_1 = 0.15$m/s and $v_2 = 2.2$m/s), which is compared with theoretical prediction model results [47]. The variation process observed in current model's spread diameters over time aligns well with experimental findings, indicating that SPH-DEM model established in this study successfully simulates droplets impacting particle surfaces.

## 4. Discussion

### 4.1 The impact velocity of droplets and its influence on droplet retraction

The SPH-DEM coupled model is employed in this section to simulate the process of droplets impacting the surface of particles at different velocities. The model parameter settings are presented in Table 2. The droplet impact velocities used were 0.38 m/s, 0.88 m/s, 1.0 m/s, and 1.48 m/s respectively. The movement process of droplets on the particle surface is

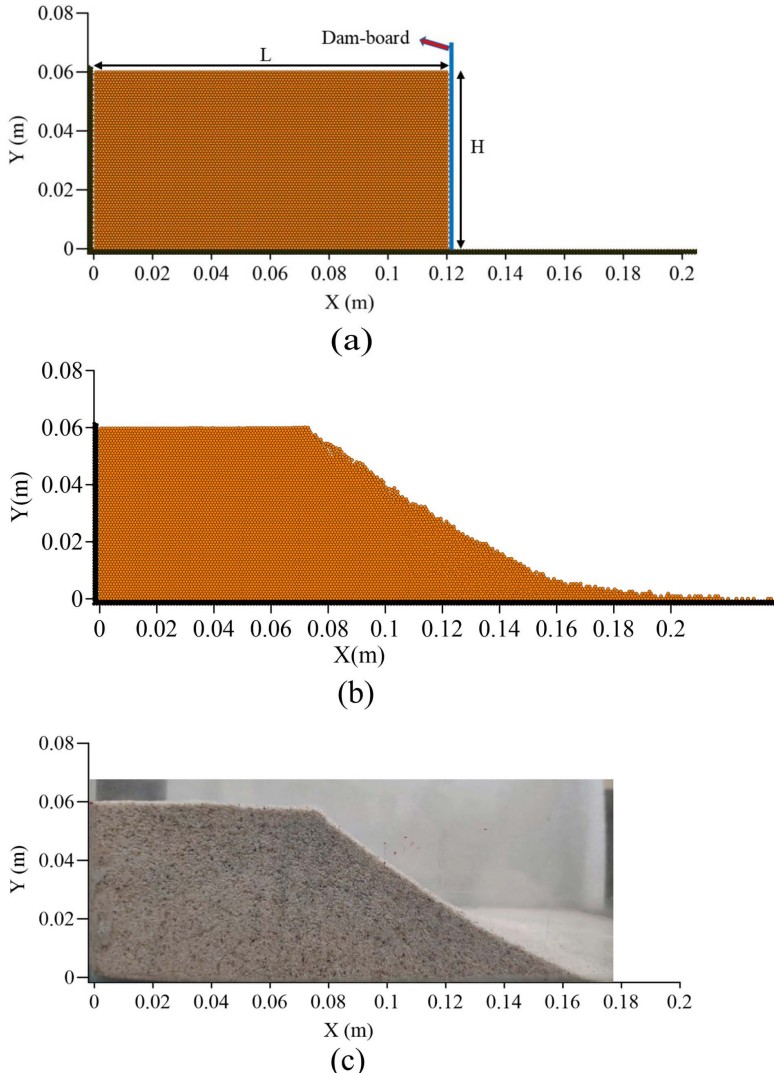

**Fig 4. (a) depicts the initial particle distribution, (b) illustrates the stable particle distribution, and (c) presents the experimental observation of the stable state of particles.**

illustrated in Fig 10, while Fig 11 demonstrates the variation of spread diameter with different impact times for droplets at various velocities.

The movement of droplets impacting the particle surface with different velocities is illustrated in Fig 10, showcasing various time intervals. The droplet velocities depicted in Fig 10(a)~(d) are 0.38m/s, 0.88m/s, 1.0m/s, and 1.48m/s respectively. This study categorizes the motion process of a droplet impacting particles into three stages: impact, spread, and retraction. During the initial stage ($t = 0$ms ~ 1ms), referred to as impact stage, droplets descend with distinct initial velocities and make contact with the particle surface resulting in an impact on the particles themselves. The droplets impact the powder surface with a specific initial velocity. During this stage, the primary form of energy possessed by the droplet is kinetic energy. Upon impact, the conversion of kinetic energy into other forms of energy commences, including deformation energy of the powder surface and internal energy within the droplet itself [48]. Compression occurs at the bottom of these droplets leading to a decrease in flow velocity while rapidly increasing pressure levels are observed simultaneously.

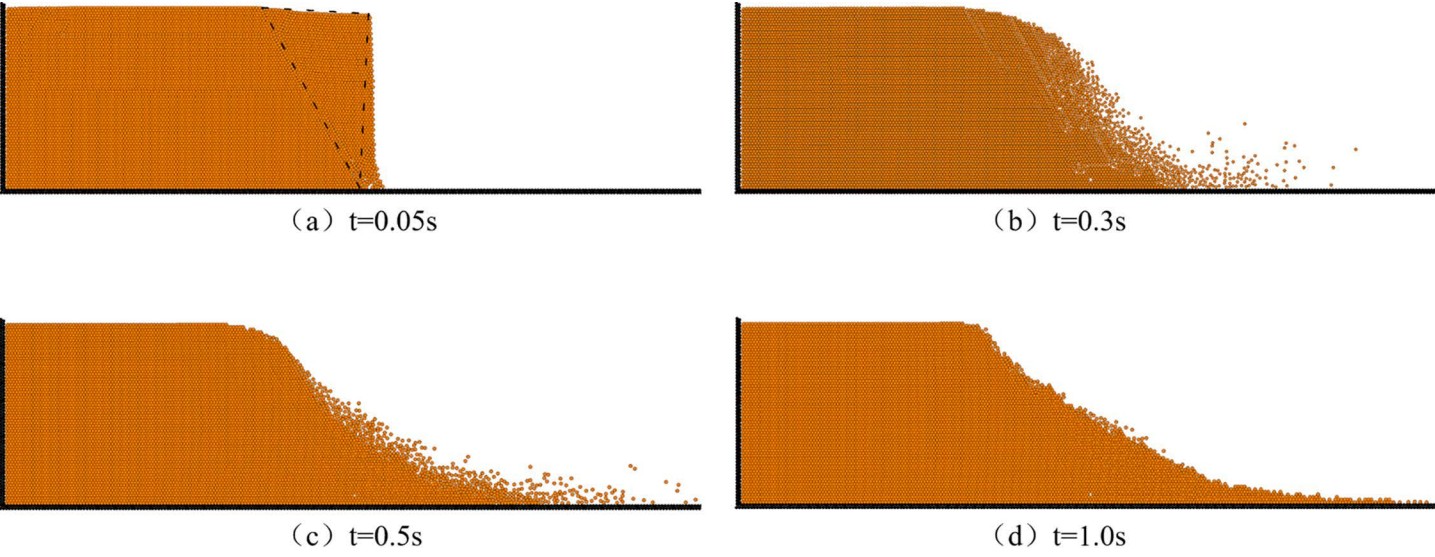

**Fig 5. The particle distribution at various time intervals.**

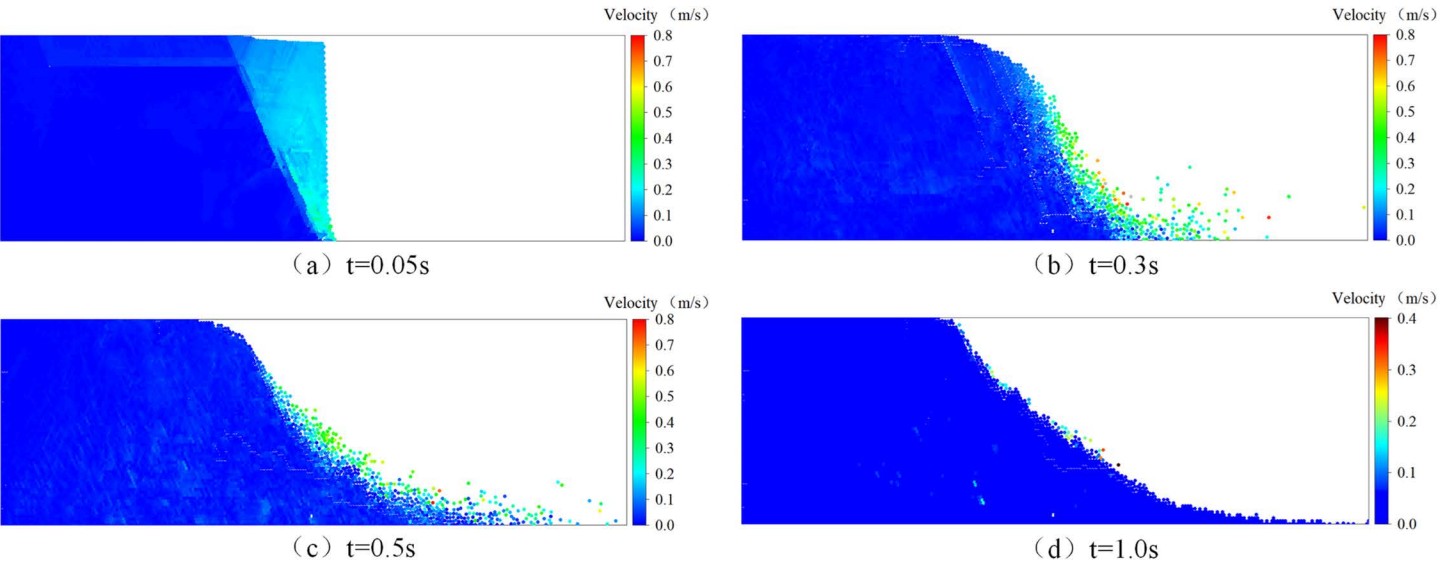

**Fig 6. Particle velocity distribution at different time.**

Particle velocity increases at both ends causing them to move towards opposite directions during this collision process without any noticeable deformation characteristics on either the droplet or particle surfaces being detected so far.

The second stage is droplet spreading. It can be observed from Fig 11 that the maximum diffusion diameter of the droplet increases with an increase in the droplet impact velocity, and the time taken for the droplet to spread to its maximum diameter also increases. Fig 10 shows that the droplets move towards both ends along the surface of the particle body, where fluid particles and solid particles interpenetrate due to gravity and capillary action. As the droplet spreads, it moves together with the particles, resulting in deformation of their surfaces. The entrainment effect reduces kinetic energy of fluid

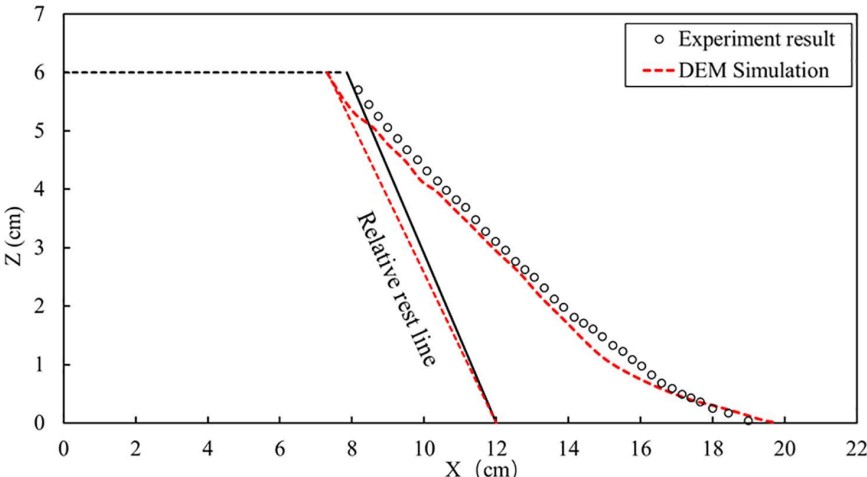

**Fig 7. Comparison of DEM simulation and experimental results of dry particle collapse.**

**Table 1. Model parameters setting information.**

| SPH parameters | Value | DEM parameters | Value |
|---|---|---|---|
| Fluid particle density (kg/m³) | 1000.0 | Particle density (kg/m³) | 3000.0 |
| Dynamic viscosity coefficient (Pa·s) | $1.0 \times 10^{-3}$ | Particle radius (μm) | 31 |
| Diameter of droplet (mm) | 2.1 | Particle elastic modulus (Pa) | $55 \times 10^{9}$ |
| Surface tension (N/m) | 0.04 | Particle shear modulus (Pa) | $22 \times 10^{9}$ |
| Droplet initial velocity (m/s) | 0.15, 2.2 | Granular Poisson's ratio | 0.25 |
| Smooth kernel function | cubic spline | Coefficient of kinetic friction | 0.4 |
| Particle Wetting Coefficient | 0.01 | Static friction coefficient | 0.8 |
| SPH time step (s) | $1.0 \times 10^{-5}$ | DEM time step(s) | $3.0 \times 10^{-6}$ |
| Gravitational acceleration (m/s²) | 9.8 | | |

particles on the contact surface, leading to a lower velocity at the lower part of spreading compared to that at upper part. Consequently, a crown-shaped formation occurs for each droplet [47].

The third stage is droplet retraction, during which the droplet begins to retract due to surface tension after spreading to its maximum diameter. Capillary action causes the particles on the contact surface of the droplet to retract along with it. Momentum exchange occurs between the fluid and solid particles wrapped around them during this process. Consequently, under identical conditions, the retraction speed of particles on the upper surface of the droplet becomes significantly more pronounced than that on its lower surface. The particles on the droplet's surface move from underneath towards the top, indicating a tendency for wrapping and accumulation. Moreover, particle thickness gradually increases as retraction progresses.

The influence of droplet impact velocity on droplet retraction time can be observed from Fig 11. A larger absolute value of k in Fig 11 indicates a faster retraction rate for the droplet. The properties of particles entrained on the under surface of the droplet have a specific impact on its retraction rate during the process. Therefore, the next section analyzes the movement of droplets on particle surfaces with different particle diameters.

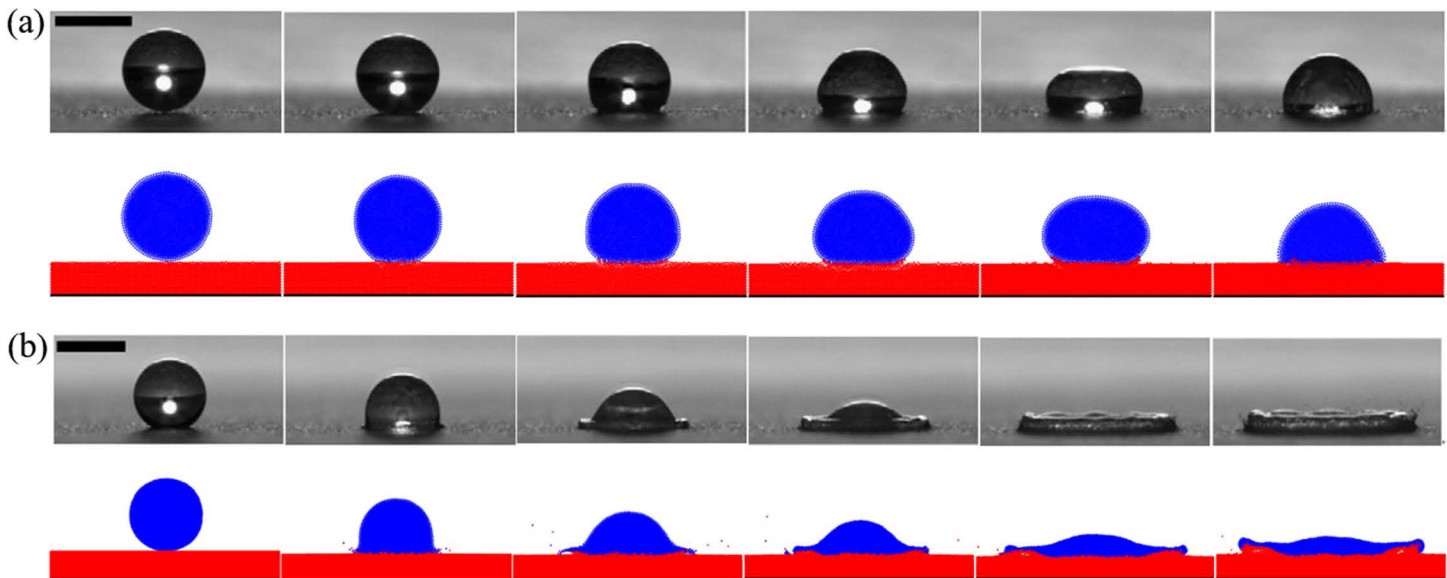

**Fig 8. Comparison of test results of droplet impacting particles surface with simulation results (a)** $v_1$ **=0.15 m/s, time t=0, 0.5, 1.5, 2.5, 4.5, and 6.5ms; (b)** $v_2$ **=2.2 m/s, the times are t=0, 0.2, 0.4, 0.6, 1.0, and 1.4ms, respectively.**

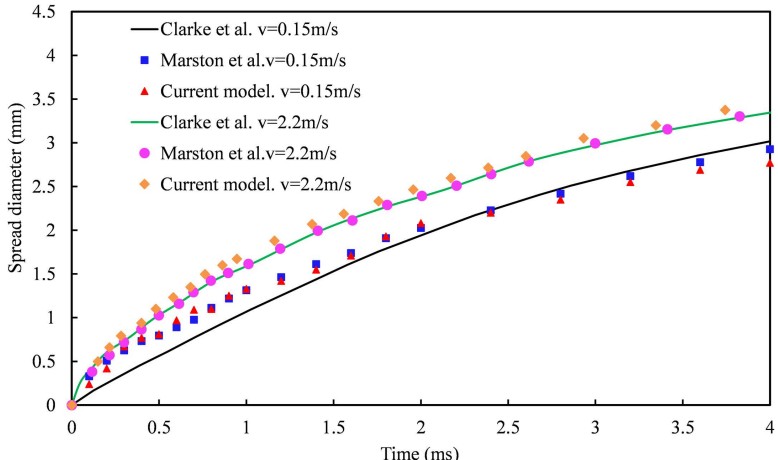

**Fig 9. The droplet velocity** $v_1$ **=0.15m/s and** $v_2$ **=2.2m/s, the relationship between the droplet spread diameter on the particles surface.**

## 4.2 The influence of bed particle size on droplet retraction

In order to investigate the influence of surface particle properties on droplet impact spread, the SPH-DEM coupled model is employed in this section to simulate droplet impacts on surfaces with varying particle diameters. The model parameter settings are presented in Table 3. Fig 12 illustrates the movement of droplets at different time intervals on surfaces with distinct particle diameters, while maintaining a constant droplet impact velocity of 1.0 m/s. Specifically, Fig 12(a)~(c) correspond to particle diameters of 70 μm, 40 μm, and 20 μm respectively. Additionally, Fig 13 demonstrates the variation process of droplet spread diameter over time on surfaces with different particle diameters.

**Table 2. Model parameters of droplet impacting particles surface with different velocities.**

| SPH parameters | Value | DEM parameters | Value |
|---|---|---|---|
| Fluid particle density(kg/m³) | 1000.0 | Particle density (kg/m³) | 3000.0 |
| Dynamic viscosity coefficient (Pa·s) | $1.0 \times 10^{-3}$ | Particle radius(μm) | 20 |
| Diameter of droplet (mm) | 2.1 | Particle Elastic Modulus/MPa | $22 \times 10^9$ |
| Surface tension (N/m) | 0.04 | Particle shear modulus/MPa | $22 \times 10^9$ |
| Droplet initial velocity (m/s) | 0.38, 0.88, 1.0, 1.48 | Granular Poisson's ratio | 0.25 |
| Smooth kernel function | cubic spline | Coefficient of kinetic friction | 0.4 |
| Particle Wetting Coefficient | 0.05 | Static friction coefficient | 0.8 |
| SPH time step (s) | $1.0 \times 10^{-5}$ | DEM time step (s) | $3.0 \times 10^{-6}$ |
| Gravitational acceleration (m/s²) | 9.8 | | |

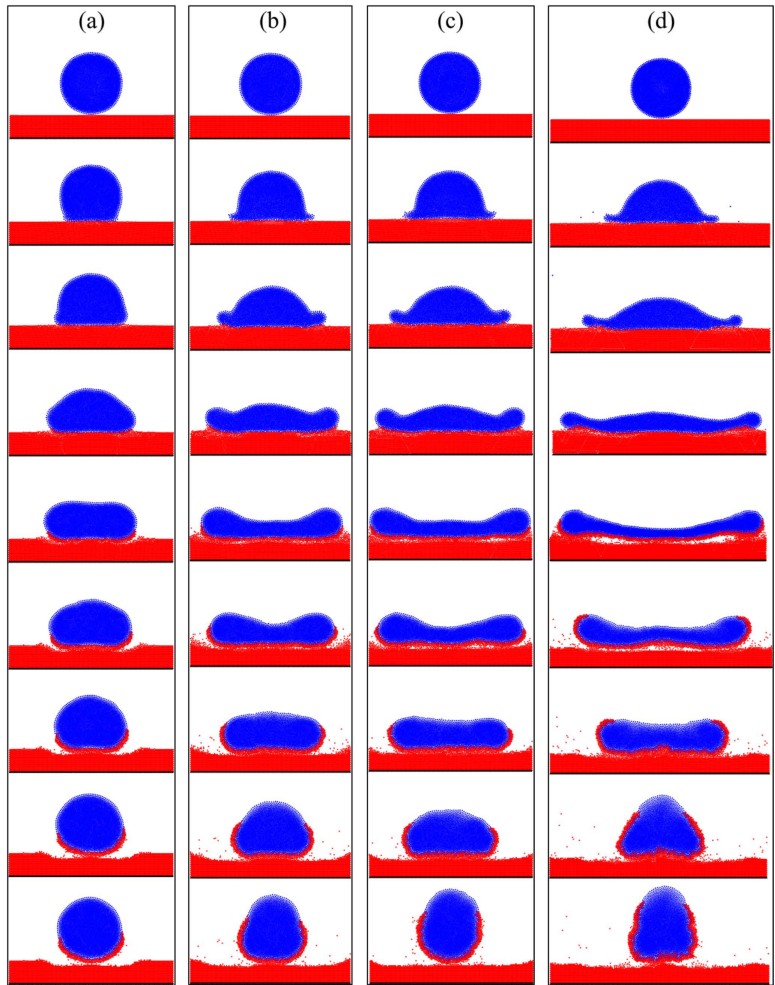

**Fig 10. Droplet spread process with different impact velocities**, (a) Droplet velocity $v_1 = 0.38$m/s, t; (b) Droplet velocity $v_2 = 0.88$m/s; (c) Droplet velocity $v_3 = 1.0$m/s; (d) Droplet velocity $v_4 = 1.48$m/s; $t = 0$, 0.5, 1.0, 2.0, 4.0, 6.0, 8, 10.0, 12.0ms, respectively.

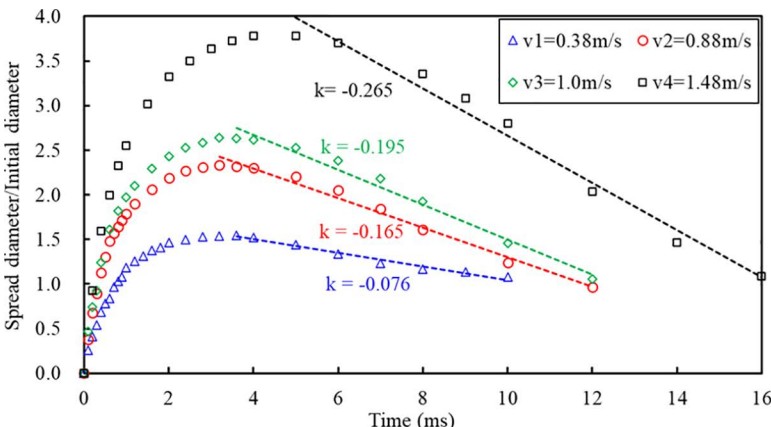

**Fig 11. Variation of the spread diameter of droplets on the surface with time at different impact velocities, and droplet retraction rate.**

It can be seen from Fig 12 that the droplets impact the particles surfaces of three different particle diameters at the same speed, and the movement process of the droplets can also be divided into three stages. The first stage, the droplet contacts and impact the surfaces. No obvious deformation of the surfaces occurred during the impact stage. In the second stage, the droplet began to spread on the surfaces, and the droplet reached maximum spreading diameter between 3.5ms-4.0ms. It can be seen that when the droplet spreads to the maximum diameter, the particles are already entrained at the bottom of the droplet. At this time, the surfaces were deformed, and erosion pits were formed under the impact of droplets. In the third stage, the droplet starts to retract under the action of surface tension. During the retraction, momentum exchange occurs between the drop and the particles. The retraction speed of the particles on the upper surface of the droplet is significantly greater than that of the lower surface, and the particles on the upper surface of the droplet retract to form bulges. The particles on the droplet's surface undergo migration from the lower to the upper surface, leading to the formation of a particle coating on its surface. The simulation results can clearly describe the nucleation process of droplets being wrapped by particles in this study. It can be seen from Fig 13 that under the same droplet impact velocity, the maximum spread diameter of the droplets on the particle diameter $d_2=40\mu m$ and $d_3=70\mu m$ particles bed is basically the same, while the maximum diffusion diameter of the droplet on the $d_1=20\mu m$ particles bed slightly smaller than the previous two. The simulation results (Fig 13) show that the droplet retraction rate is the largest on the $d_1=20\mu m$ particles bed, followed by the $d_2=40\mu m$ particles bed, and finally 70μm. Therefore, it can be concluded that the particle diameter has a particular influence on the droplet retraction. The rate of retraction of the droplets slows down as the particle diameter increases.

## 5. Conclusion

This study presents an SPH-DEM coupling model, which considers the viscous force between droplets and particles as well as the capillary force generated by the voids among particles. It is employed to investigate the interaction between droplets and powder particles. The following conclusions have been drawn:

(1) The correctness of the SPH method for simulating the movement process of droplets in this paper is verified through the square droplet oscillation test and the simulation of droplet impact on hydrophobic surfaces.

(2) The accuracy of the established DEM model in this paper for studying the movement process of particles is validated by the particle collapse experiment.

(3) The SPH-DEM coupling model is utilized to simulate the experiment of droplet impact on powder surfaces. The spreading process of the droplet is in good agreement with the experimental results, demonstrating that the established model in this paper can be applied to the research of droplet impact on powder surfaces.

**Table 3. Droplet impact particles surface model parameter settings with different particle diameters.**

| SPH parameters | Value | DEM parameters | Value |
|---|---|---|---|
| Fluid particle density/(kg/m³) | 1000.0 | Particle density (kg/m³) | 3000.0 |
| Dynamic viscosity coefficient (Pa·s) | $1.0 \times 10^{-3}$ | Particle radius(μm) | 10, 20, 35 |
| Diameter of droplet (mm) | 2.1 | Particle elastic modulus (MPa) | $22 \times 10^9$ |
| Surface tension (N/m) | 0.04 | Particle shear modulus (MPa) | $22 \times 10^9$ |
| Droplet initial velocity (m/s) | 1 | Granular Poisson's ratio | 0.25 |
| Smooth kernel function | cubic spline | Coefficient of kinetic friction | 0.4 |
| Particle Wetting Coefficient | 0.05 | Static friction coefficient | 0.8 |
| SPH time step (s) | $1.0 \times 10^{-5}$ | DEM time step (s) | $3.0 \times 10^{-6}$ |
| Gravitational acceleration (m/s²) | 9.8 | | |

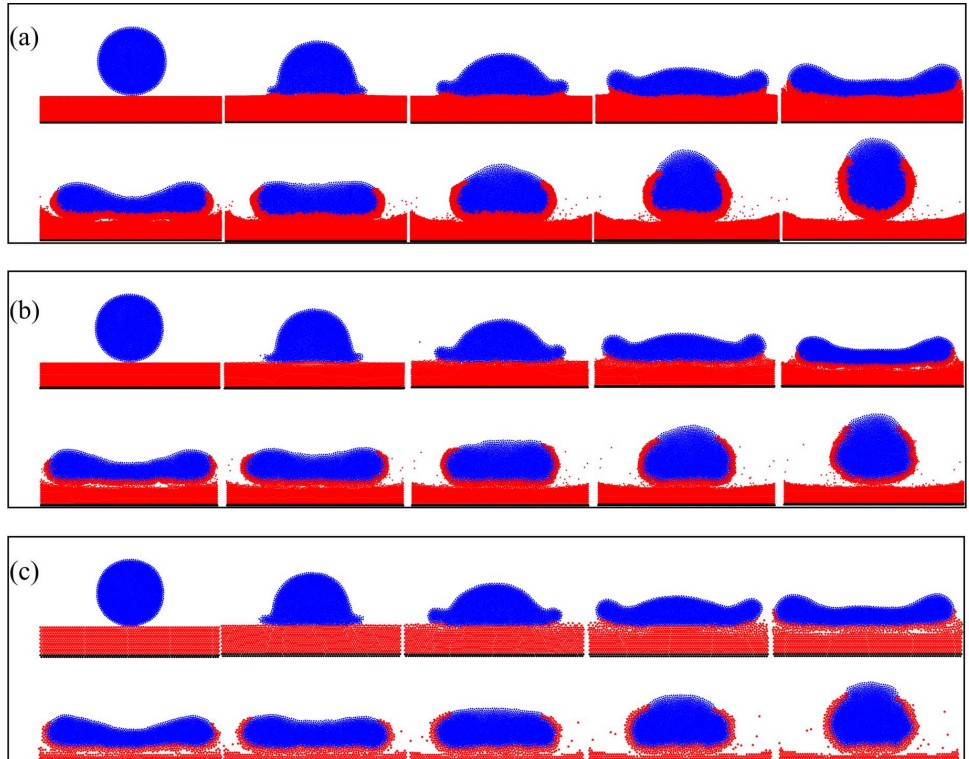

**Fig 12. Movement process of droplets on the particle surface,** (a) particle diameter $d_1$ = 20μm; (b) particle diameter $d_2$ = 40μm; (c) particle diameter $d_3$ = 70μm; $t$ = 0, 0.5, 1.0, 2.0, 4.0, 6.0, 8,.0 10.0, 12.0, and 15.0ms, respectively.

(4) Furthermore, the influence of the impact velocity of droplets and the diameter difference of powder bed particles on the rebound rate of droplets after impacting the powder bed surface is studied through the SPH-DEM model. The results indicate that the greater the impact velocity of droplets, the faster the rebound rate. The smaller the particle diameter, the faster the rebound rate.

The numerical simulation approach in this paper will offer new possibilities for in-depth exploration of the micro-mechanical mechanism of the interaction between droplets and powder particles.

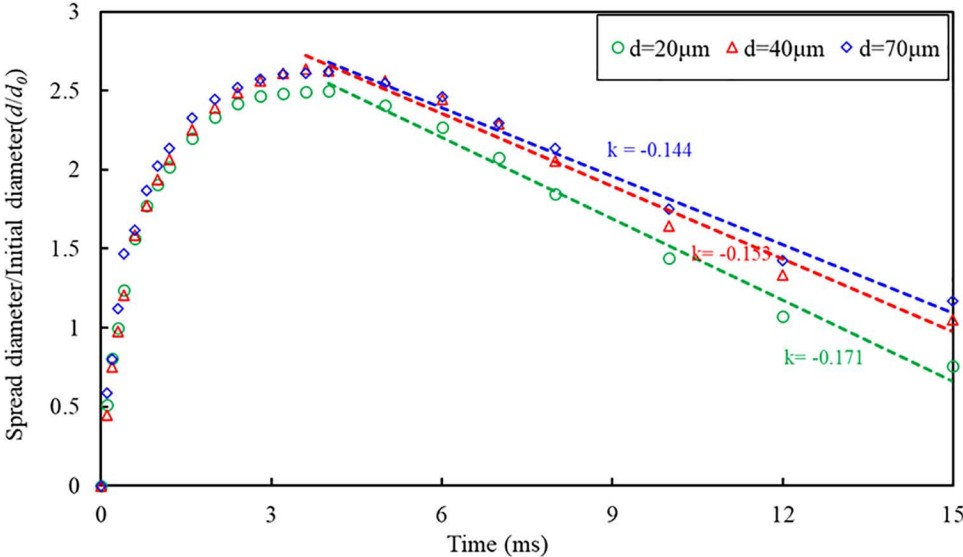

**Fig 13. The time-dependent process of the droplet spread diameter on the particles surface with different particle diameters, and the droplet retraction rate.**

## Author contributions

**Conceptualization:** Shilong Bu.

**Data curation:** Shilong Bu.

**Methodology:** Daming Li.

**Resources:** Hu Tao.

**Validation:** Shilong Bu.

**Writing – original draft:** Shilong Bu.

**Writing – review & editing:** Wenjie Hou.

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
