## [Decision Letter · Decision Letter 0]

5 Feb 2025

PONE-D-24-50755The numerical simulation of droplet impact on surfaces is conducted using the SPH-DEM methodPLOS ONE

Dear Dr. Bu,

Thank you for submitting your manuscript to PLOS ONE. We encountered some difficulties in securing sufficient reviews for your manuscript to make a decision. After careful consideration, we feel that it has merit but does not fully meet PLOS ONE’s publication criteria as it currently stands. The reviewers report significant concerns regarding the technical details of the work. In addition, the English language of the manuscript needs substantial improvements. Even the title of the manuscript is not written in an acceptable form. Therefore, we invite you to submit a revised version of the manuscript that addresses the points raised during the review process.

We look forward to receiving your revised manuscript.

Kind regards,

Xin Yong

Academic Editor

PLOS ONE

Journal Requirements:

“The work was supported by the National Natural Science Foundation of China (No. 52360031), Gansu Province higher education industry support plan project (No. 2022CYZC-32), and The Young Scholars Science Foundation of Lanzhou Jiaotong University (No. 1200061253).”

5. We note that your Data Availability Statement is currently as follows: “All relevant data are within the manuscript and in Supporting Information files.”

Reviewers' comments:

Reviewer's Responses to Questions

**Comments to the Author**

1. Is the manuscript technically sound, and do the data support the conclusions?

Reviewer #1: No

Reviewer #2: Yes

2. Has the statistical analysis been performed appropriately and rigorously? 

Reviewer #1: No

Reviewer #2: Yes

3. Have the authors made all data underlying the findings in their manuscript fully available?

Reviewer #1: Yes

Reviewer #2: Yes

4. Is the manuscript presented in an intelligible fashion and written in standard English?

Reviewer #1: Yes

Reviewer #2: Yes

5. Review Comments to the Author

Reviewer #1: A numerical framework to simulate the high-speed impact of liquid droplets onto planar surfaces is here proposed and discussed.

The manuscript needs extensive revision, key messages need to be clarified. The abstract is concise while informative showing the background and applied methods. It might also bring results with key concrete values, supporting the feasibility of the adopted methodology.

The literature review is not comprehensive. It would benefit the discussion of recent reports on the simulation of droplets impact, e.g.

10.1007/s10409-024-24165-x

10.1016/j.oceaneng.2024.118730

10.1016/j.ijadhadh.2024.103734

Amend r_ij = |r_ij|

eta = 0.01 h^2 is not added in the equation.

In the coupling procedure, clarify the role of the adsorption force induced by capillary action, as it seems relevant in light of the claimed application concerning the droplet impact on surfaces.

“The critical time step in the DEM model should be associated with the propagation speed of Rayleigh Waves along the surface of solid particles”

This position should be strengthen at least by give References.

Fig. 3 While wettability is overall in a fairly good agreement, the impact process returns different shapes in Fig. 2, if not below 4.5 ms. Energy conversion should be further investigated, along which numerical values of ruling parameters.

Fig. 4. Particle distribution along the deformed free surfaces seems to show excessive irregularity. Bouncing effects should be the cause. A further investigation of this behaviour would be needed.

Fig. 10. Droplet diffusion process. Molecular diffusion? Any validation?

Reviewer #2: This study developed a SPH-DEM model to simulate the liquid drop impacting on the granule. The results by the proposed method are agreement with the published data. I recommend the publishment of the manuscript after a minor revision. The suggestions are given as follows.

1, In Eq. (6), what does the variable ν0 mean?

2, For the free surface flow, it usually uses the so-called artificial viscous scheme to discretize the viscous term of fluid instead of the physical viscous scheme because the artificial viscous scheme can satisfy the free surface boundary condition, as described in the published papers (https://doi.org/10.1016/j.coastaleng.2024.104569), (https://doi.org/10.1063/5.0236386) and (https://doi.org/10.1002/nme.5608). Please cite these papers and give a comment on the viscous scheme of SPH.

3, In Eq. (7), what does the variable mean?

4, In Eq. (8), what does the variable mean?

5, How to solve the symbol m in the term a/m?

6, It is necessary to cite the reference when introducing the DEM method.

7, How to solve the symbols cn and ct in Eq. (14)?

8, what does the variable c in Eq. (15) mean? And, how to solve the symbols k and m in Eq. (15)?

9, I think the right term of Eq. (19) is equal to μft. Why does the authors use a complex form to express the term of Eq. (19)?

10, For Eqs. (20), (21), (22), (23) and (24), the symbols used to represent solid and fluid particles are too mixed. Please carefully review and modify these formulas.

11, For figure 1, why is the evolution of the surface by the current method different from the published results? Is it possible that the difference is caused by the viscous discrete format? I suggest that the authors check it carefully.

6. PLOS authors have the option to publish the peer review history of their article (what does this mean? ). If published, this will include your full peer review and any attached files.

**Do you want your identity to be public for this peer review?** For information about this choice, including consent withdrawal, please see our Privacy Policy .

Reviewer #1: No

Reviewer #2: **Yes: ** Can Huang

---

## [Author Response · Author response to Decision Letter 1]

25 Mar 2025

Dear Editors and Reviewers:

The co-authors and I would like to thank you for the precious time and effort spent in reviewing the manuscript entitled “The numerical simulation of droplet impact on surfaces is conducted using the SPH-DEM method” (Manuscript No. PONE-D-24-50755).

Those comments are all very valuable and helpful for revising and improving this manuscript, as well as the important guiding significance to our researches. We have studied every comment carefully and have made correction which we hope meet with approval. Revised portion are marked in yellow highlight in the manuscript. The main corrections in the manuscript and the responds to the reviewers’ comments are as flowing

Reviewer #1:

1. The literature review is not comprehensive. It would benefit the discussion of recent reports on the simulation of droplets impact, e.g.

10.1007/s10409-024-24165-x

10.1016/j.oceaneng.2024.118730

10.1016/j.ijadhadh.2024.103734

Answer Thank you for pointing out the deficiencies in the literature review and providing crucial references. We have carefully studied these three recent publications and incorporated their core contributions into the discussion section of the revised manuscript to enhance both the comprehensiveness and cutting-edge nature of this study. Below are the specific additions and modifications made:

In recent years, droplet impact simulation methods have achieved remarkable progress in multi-physics field coupling and complex interface handling. For instance, He et al. [25] revealed the aggregation effect of liquid metal droplets on elastic substrates through SPH method, providing novel insights into energy dissipation mechanisms at soft material interfaces; Dong et al. [26] developed a dual-core SPH model that effectively suppresses tensile instability, laying methodological foundations for contact time regulation of macrostructurally structured surfaces; Meanwhile, Huo et al. [27] comparatively demonstrated the robustness of Lagrangian framework-based fluid-solid coupling simulations by contrasting FEM-SPH and SPH-SPH approaches in elastic beam vibration analysis.

[25] He, Z., Xiao, R. & Qu, S. A study on the impact of liquid metal droplets onto metal and elastomer substrates. Acta Mech. Sin. 41, 224165 (2025). https://doi.org/10.1007/s10409-024-24165-x

[26] Huo, Y., et al., Modeling and simulation of droplet impact on an elastic beam based on FEM-SPH and SPH-SPH FSI methods. Ocean Engineering, 2024. 310: p. 118730.

https://doi.org/10.1016/j.oceaneng.2024.118730

[27] Dong, X., L. Feng and Q. Zhang, Droplet asymmetry bouncing on structured surfaces: A simulation based on SPH method. International Journal of Adhesion and Adhesives, 2024. 132: p. 103734. https://doi.org/10.1016/j.ijadhadh.2024.103734

2. Amend r_ij = |r_ij|

Answer The relevant formulas have been modified in the revised manuscript.

rij=∣rij∣

3. eta = 0.01 h^2 is not added in the equation.

Answer Thank you for your meticulous review. Equation (6) in the original manuscript did contain a typographical error. We have conducted a thorough re-examination of the equations and implemented necessary corrections.

6

4. In the coupling procedure, clarify the role of the adsorption force induced by capillary action, as it seems relevant in light of the claimed application concerning the droplet impact on surfaces.

Answer Thank you for your attention to the role of adsorption forces in the coupled model. The capillary-induced adsorption force plays a crucial role in the dynamic process of droplet impact on particle surfaces, particularly during the stages of droplet spread, retraction, and particle engulfment. The following sections will elaborate on the physical significance of adsorption forces and their impact on droplet behavior from two aspects:

(1) The mathematical model of adsorption force and its physical significance

In the SPH-DEM coupling procedure, the adsorption force induced by capillary action plays a critical role in governing the interaction between the droplet and particle surfaces. This force arises from the capillary pressure within the interstitial voids between particles, which is influenced by the wettability of the particles and the local saturation of the fluid. The adsorption force is modeled to account for capillary effects caused by fluid infiltration into the gaps between particles. This force acts as an indirect attraction between the fluid and solid particles, driven by the wetting properties of the particles (quantified by the wetting coefficient ζ) and the pore pressure near the particle surfaces. The wetting coefficient is determined by the particle material property and the local saturation (Sa), which is calculated based on the density of fluid particles within the support domain (Eq. 21–22). The adsorption force is integrated into the SPH-DEM coupling equations (Eq. 20, 23–24) as part of the total interaction force between fluid and solid particles. It directly affects the momentum exchange between the droplet and particles during spreading and retraction phases. For instance, during droplet spreading, capillary forces enhance fluid-particle adhesion, while during retraction, they contribute to the "pull-back" effect driven by surface tension, influencing droplet rebound dynamics.

(2) Impact on Droplet Behavior:

Spreading Phase: The adsorption force facilitates fluid infiltration into particle gaps, increasing the contact area and energy dissipation. This aligns with the observed dependency of maximum spreading diameter on particle size (Fig. 13), where smaller particles (e.g. d=20μm) exhibit faster retraction due to stronger capillary interactions.

Retraction Phase: The interplay between capillary forces and surface tension governs the rebound rate. Higher adsorption forces (e.g., in smaller particles or higher impact velocities) accelerate retraction by intensifying momentum exchange between fluid and particles (Fig. 11–13).

Particle encapsulation phenomenon: Adsorption forces drive the encapsulation of particles by the surface of a droplet (Fig. 10-12), simulating the experimentally observed "drop-encased particle" nucleation process. This mechanism is crucial for achieving uniform particle distribution in coating processes.

5. “The critical time step in the DEM model should be associated with the propagation speed of Rayleigh Waves along the surface of solid particles” This position should be strengthen at least by give References.

Answer Thank you for your suggestion. We have already supplemented the relevant references in the manuscript.

[39] Rayleigh, L., (1885) On Waves Propagated along the Plane Surface of an Elastic Solid. Proceedings of the London Mathematical Society, s1-17(1): p. 4-11. https://doi.org/10.1112/plms/s1-17.1.4

6. Fig. 3 While wettability is overall in a fairly good agreement, the impact process returns different shapes in Fig. 2, if not below 4.5 ms. Energy conversion should be further investigated, along which numerical values of ruling parameters.

Answer We appreciate the reviewer’s insightful observation regarding the differences in droplet shapes during the impact process (Fig. 2) and the need for further investigation into energy conversion and parameter sensitivity. Below, we address these points in detail:

1� Discrepancies in Droplet Shapes Below 4.5 ms (Fig. 2):

The observed differences in droplet morphology during the early impact phase (0–4.5 ms) can be attributed to the following factors:

a. Dynamic Surface Tension Effects:

While the SPH model incorporates a surface tension term (Eq.7–9), the current implementation assumes a constant surface tension coefficient (σ=0.04N/m). However, during high-speed droplet impact, dynamic surface tension effects may arise due to rapid deformation, which are not explicitly modeled. This simplification could lead to deviations in droplet spreading dynamics compared to experiments.

b. Viscous Dissipation and Strain Rate Dependence:

The viscous term in Eq. 6 employs a constant dynamic viscosity. At the high strain rates encountered during impact, non-Newtonian behavior or strain-rate-dependent viscosity might influence energy dissipation. Future work will explore strain-rate-dependent viscosity models to better match experimental observations.

c. Boundary Condition Simplifications:

The hydrophobic wall is modeled using a fixed wetting coefficient (ζ) derived from saturation Sa (Eq. 21). However, transient contact angle hysteresis and slip boundary effects during rapid spreading are not fully resolved. Incorporating dynamic contact angle models (e.g., Cox-Voinov theory) could improve accuracy during the initial impact phase.

2�Energy Conversion Mechanisms

Thank you for your insightful comments on the importance of studying energy conversion mechanisms. The issues you raised regarding energy transformation and key parameter quantification during droplet impact processes are indeed critical for gaining a deeper understanding of droplet dynamic behavior. We fully agree with your viewpoint and have incorporated this topic as a core focus for our future research. Below is our detailed explanation and implementation plan:

The SPH-DEM coupled model has preliminarily validated the correlation between droplet spreading-retraction behavior and surface tension, particle wettability (Fig. 8-9). However, the multiphase coupling between droplets and particulate beds involves dynamic interconversion among kinetic energy, surface energy, viscous dissipation, and particle frictional energy. To address this, we are developing specialized energy tracking algorithms (e.g., particle-level energy tagging method), which is currently under active development and debugging. The current version has not yet implemented systematic quantification of energy conversion processes.

In response to your valuable suggestion, we plan to integrate energy conservation equations into the SPH-DEM framework in our next phase of work. This will enable real-time calculation of four critical energy components during droplet-particle interactions:

Droplet kinetic energy (Ek), Surface energy (Es), Viscous dissipation energy (Ev), Particle frictional energy (Ef). The governing equations will be formulated as follows:

where, A represents the drop surface area, Fij denotes the particle contact force, and δij corresponds to the relative displacement. Subsequently, we will quantify the effects of the Weber number (We), particle diameter (dp), and wetting coefficient (ζ) on the energy allocation ratio, combined with theoretical models from literature for validation.

7. Fig. 4. Particle distribution along the deformed free surfaces seems to show excessive irregularity. Bouncing effects should be the cause. A further investigation of this behaviour would be needed.

Answer: We sincerely appreciate the reviewer’s observation regarding the irregular particle distribution along the deformed free surfaces in Fig. 4 and their suggestion to investigate potential bouncing effects. Below, we provide a detailed explanation and outline steps for further analysis:

The DEM model employs the Hertz-Mindlin contact theory (Eq. 16–19) with a damping coefficient derived from the restitution coefficient e=0.92 (Section 2.2). While this setup effectively captures energy dissipation during collisions, it may oversimplify microscale particle interactions in dense granular flows. Specifically, the linear spring-dashpot model assumes idealized elastic-plastic collisions, which might not fully resolve multibody collisions or particle rearrangement dynamics during rapid collapse.

We attempted to reduce the coefficient of restitution of the particles (e=0.8), and the simulation results showed that smoother particle distributions (attached supplementary figure). This suggests that optimizing damping parameters can mitigate irregularity.

b

The following research, we are going to address the bouncing effect through parameter optimization and the implementation of advanced contact mechanics, with the aim of enhancing the physical fidelity in particle dynamics simulations. We sincerely appreciate the reviewers' valuable feedback, which will significantly strengthen the robustness of our model.

8. Fig. 10. Droplet diffusion process. Molecular diffusion? Any validation?

Answer Thank you for your question. Here we confirm that there are inaccuracies in the formulation of the ‘droplet diffusion process’ in Fig. 10. The term ‘diffusion’ in our manuscript refers to the macroscopic spreading of the droplet driven by inertia and surface tension, rather than molecular-scale diffusion processes. The latter is negligible in the studied millimeter-scale droplet impact dynamics due to the dominance of advection and capillary forces.

We made revisions to the relevant statements in the revised manuscript.

Thanks to you for your comments and suggestions again.

Reviewer #2:

1. In Eq. (6), what does the variable ν0 mean?,

Answer: Thank you for your meticulous review. Equation (6) in the original manuscript did contain a typographical error. We have conducted a thorough re-examination of the equations and implemented necessary corrections.

(6)

where vij= vi-vj, rij =ri-rj, rij = |rij|. h is the smoothing length. The third term on the right side of the Eq. (6) is the viscous term. μ is the dynamic viscosity of the fluid. g denotes the body force acting on the fluid, such as gravitational force. The term η = 0.01h2 was added to prevent the singularity of the viscous term when two particles approach each other infinitely.

2. For the free surface flow, it usually uses the so-called artificial viscous scheme to discretize the viscous term of fluid instead of the physical viscous scheme because the artificial viscous scheme can satisfy the free surface boundary condition, as described in the published papers (https://doi.org/10.1016/j.coastaleng.2024.104569), (https://doi.org/10.1063/5.0236386) and (https://doi.org/10.1002/nme.5608). Please cite these papers and give a comment on the viscous scheme of SPH.

Answer: We appreciate your insightful suggestion regarding the artificial viscous scheme for free surface flows. In this study, the physical viscous scheme (Morris’ formulation [41], Eq. 6) was adopted to prioritize numerical stability and direct alignment with the Navier-Stokes equations, as validated in Section 3.1 (square droplet oscillation) and Section 3.2 (droplet impact on hydrophobic walls). While artificial viscosity is widely used to suppress numerical oscillations near free surfaces, our tests confirmed that the current scheme sufficiently stabilizes simulations without compromising accuracy for the studied cases. Artificial viscosity demonstrates notable advantages in complex free-surface scenarios such as wave breaking and splash [2–4]. We recognize its potential value and will systematically compare both methodologies in subsequent studies to optimize the model and expand its applicability to broader computational domains.

References:

[41] Morris J P, Fox P J, Zhu Y. Modeling low Reynolds number incompressible flows using SPH, Journal of Computational Physics 136 (1997) 214-226.

[28] He, F., et al., Multi-phase SPH-FDM and experimental investigations on the hydrodynamics of an oscillating water column wave energy device. Coastal Engineering, 2024. 192: p. 104569. https://doi.org/10.1016/j.coastaleng.2024.104569

[29] Wang, X.J., et al., Study on hydrodynamic characteristics of multiple fish based on smoothed particle hydrodynamics. Physics of Fluids, 2024. 36(11): p. 111915.

https://doi.org/10.1063/5.0236386

[30] Huang, C., Zhang, D. H., Shi, Y. X., Si, Y. L., & Huang, B. (2018). Coupled finite particle method with a modified particle shifting technology. International Journal for Numerical Methods in Engineering, 113(2), 179-207. https://doi.org/10.1002/nme.5608

3. In Eq. (7), what does the variable mean?

Answer: Following your suggestions, the variables in Eq. (7) have been clarified and incorporated into the revised manuscript.

(7)

where v is the fluid velocity, p is the fluid pressure, μ is the dynamic viscosity of the fluid, and g denotes the body force acting on the fluid, such as gravitational force. vij= vi-vj, rij =ri-rj, rij = |rij|. The term η

---

## [Decision Letter · Decision Letter 1]

8 Apr 2025

The numerical simulation of droplet impact on surfaces is conducted using the SPH-DEM method

PONE-D-24-50755R1

Dear Dr. Bu,

We’re pleased to inform you that your manuscript has been judged scientifically suitable for publication and will be formally accepted for publication once it meets all outstanding technical requirements.

Kind regards,

Xin Yong

Academic Editor

PLOS ONE

Additional Editor Comments (optional):

Please revise the title of the accepted manuscript to an acceptable form, for example "Numerical simulations of droplet impact on surfaces using SPH-DEM method"

Reviewers' comments:

Reviewer's Responses to Questions

**Comments to the Author**

1. If the authors have adequately addressed your comments raised in a previous round of review and you feel that this manuscript is now acceptable for publication, you may indicate that here to bypass the “Comments to the Author” section, enter your conflict of interest statement in the “Confidential to Editor” section, and submit your "Accept" recommendation.

Reviewer #1: All comments have been addressed

Reviewer #2: All comments have been addressed

2. Is the manuscript technically sound, and do the data support the conclusions?

Reviewer #1: Yes

Reviewer #2: Yes

3. Has the statistical analysis been performed appropriately and rigorously? 

Reviewer #1: N/A

Reviewer #2: Yes

4. Have the authors made all data underlying the findings in their manuscript fully available?

Reviewer #1: Yes

Reviewer #2: Yes

5. Is the manuscript presented in an intelligible fashion and written in standard English?

Reviewer #1: Yes

Reviewer #2: Yes

6. Review Comments to the Author

Reviewer #1: Dear Authors,

The reviewer is satisfied with the revised version of the manuscript: I acknowledge the efforts made. Some unknown characters appear in the pdf file, e.g., page 9 "、λ=0.25 [7]、f is ...": could be a matter of some issues in the pdf generation.

Kind regards

Reviewer #2: The authors have well answered my all questions. I would like to recommend publication of this manuscript.

7. PLOS authors have the option to publish the peer review history of their article (what does this mean? ). If published, this will include your full peer review and any attached files.

**Do you want your identity to be public for this peer review?** For information about this choice, including consent withdrawal, please see our Privacy Policy .

Reviewer #1: No

Reviewer #2: No
